# Rational Design and Synthesis of AT1R Antagonists

**DOI:** 10.3390/molecules26102927

**Published:** 2021-05-14

**Authors:** Nikitas Georgiou, Vasileios K. Gkalpinos, Spyridon D. Katsakos, Stamatia Vassiliou, Andreas G. Tzakos, Thomas Mavromoustakos

**Affiliations:** 1Department of Chemistry, National and Kapodistrian University of Athens, Panepistimiopolis Zografou, 15771 Athens, Greece; nikitasgalleti93@hotmail.com; 2Department of Chemistry, Section of Organic Chemistry and Biochemistry, University of Ioannina, 45110 Ioannina, Greece; v.galpinos@uoi.gr (V.K.G.); katsakos.spiros@gmail.com (S.D.K.); 3University Research Center of Ioannina (URCI), Institute of Materials Science and Computing, 45110 Ioannina, Greece

**Keywords:** sartans, hypertension, AT1R, Angiotensin II, rational design

## Abstract

Hypertension is one of the most common diseases nowadays and is still the major cause of premature death despite of the continuous discovery of novel therapeutics. The discovery of the Renin Angiotensin System (RAS) unveiled a path to develop efficient drugs to fruitfully combat hypertension. Several compounds that prevent the Angiotensin II hormone from binding and activating the AT1R, named sartans, have been developed. Herein, we report a comprehensive review of the synthetic paths followed for the development of different sartans since the discovery of the first sartan, Losartan.

## 1. Introduction

Hypertension is a medical condition where the blood pressure in the arteries is elevated. However, even if one does not exhibit symptoms, long-term high blood pressure is the major risk factor for numerous pathologies, such as stroke or heart failure. Angiotensin II (AII) receptor blockers are a class of molecules that act in the Renin Angiotensin System (RAS) through binding to the AT1 receptor (Angiotensin II receptor type 1, AT1R) and preventing its activity. AT1R is a G-protein-coupled receptor that is responsible for AII pathophysiological actions [1].

Sartans are a class of therapeutics that act in RAS through AII binding to the AT1R. The first of this class of molecules was losartan, which was approved by the FDA in 1995, and it was released to the market in April 1995. Specifically, Duncia et al. published the discovery of DuP753 (losartan), a potent, orally active nonpeptide Angiotensin II (AII) receptor antagonist (see Figure 1) and other derivatives. Their aim was to develop a peptide-mimetic molecule based on structural features of the AII peptide, which are responsible for its biological activity. This is a similar approach to that followed by Cushman and Ondetti that led to the discovery of captopril [2,3].

The first, important, evidence that triggered the discovery of losartan was the fact that the C-terminal amino acids of AII are responsible for binding to the AT1R. The C-terminal tetrapeptide (AII 5–8) has a very low binding affinity to the AT1R, but at high concentration displaces [^125^I]-AII from the receptor. The first hypothesis was that both AII and the lead molecules of the Takeda Pharmaceutical Company Limited (imidazole-5-acetic acid derivatives) [4], bind to the same receptor site. They hypothesized that the poor binding affinity of the Takeda’s molecules was due to their small size and partial resemblance to the AII. They overlapped compound **3** (Figure 1) with a conformer of AII in the following way: (a) the carboxylic acid group of **3** was aligned with the C-terminal carboxylic acid of AII; (b) the imidazole of **3** was aligned with the imidazole His6 of AII; (c) the lipophilic n-butyl side chain of **3** was pointed into an area where the Ile5 residue’s aliphatic side chain resides; (d) the benzyl group was pointing toward the N terminus of AII. The para position of the benzyl group seemed to be the ideal place to attach a functional group to enlarge these compounds and allow them to better mimic AII. As AII contains two acidic groups, the β-COOH of Asp and the OH of Tyr they introduced a second acidic functional group, namely, a carboxylic acid into the para position of the phenyl ring of **3**. Losartans’ active metabolite is EXP 3174 that is responsible for the mapped antihypertensive activity [5,6]. 

Losartan is commercially available and it was the first orally active drug, approved by the FDA for the treatment of hypertension. The production of losartan was followed by seven other commercially available derivatives that were classified as sartans. It is of interest to note that losartan was designed and developed based solely on the solution conformation of the peptide hormone AII which was further modified by rational design steps. At the time there were no crystallographic data about its structure, which had yet to be crystallized [7] as also of the AT1R. Although numerous conformations have been published in different environments for AII [5,6,7,8,9,10,11,12,13,14,15,16,17,18] the putative bioactive conformation of AII has been recently unveiled through its binding to the AT1R [7,10,19,20]. In this article we will focus on the major synthetic routes followed to commercially available sartans (Figure 2).
Figure 1Design efforts that led to the development of losartan.
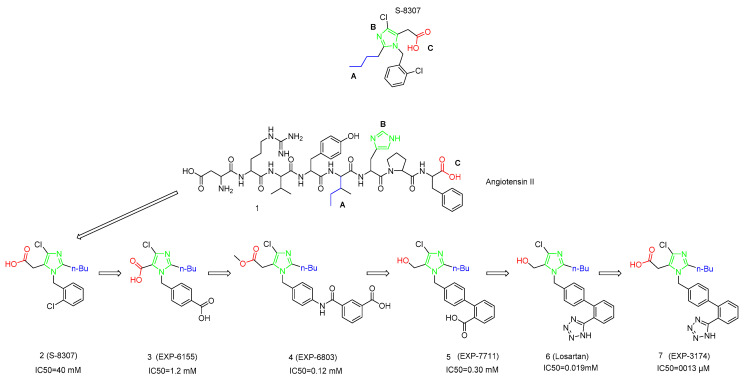

Figure 2Structures of commercially available sartans.
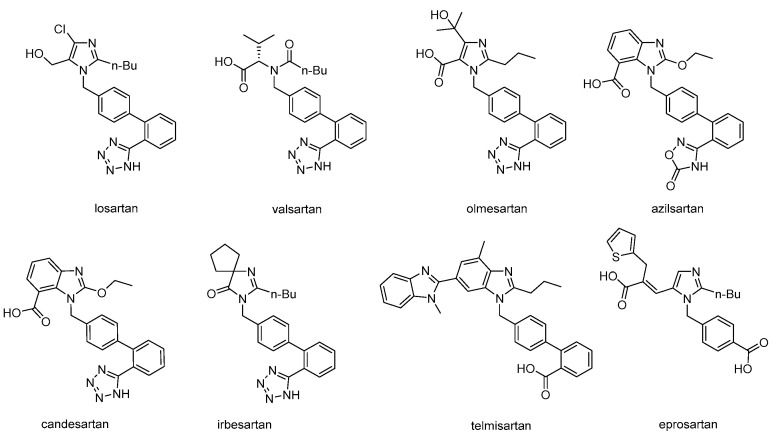


## 2. Synthetic Routes of Different Sartans

### 2.1. Synthetic Routes of Losartan

One common structural characteristic of the commercially AT1R antagonists is that they contain an acidic group such as carboxylate or tetrazole or 5-oxo-1,2,4-oxadiazole ring. Tetrazole and 5-oxo-1,2,4-oxadiazole ring groups are prepared from nitrile group and are bioisosteric. They also retain a more favorable pharmacological effect, pharmacokinetic profile and they are metabolically more stable compared to carboxylate group. The replacement of a functional group by another that modifies beneficially the pharmaceutical profile of a compound is a fundamental medicinal chemistry strategy known as isosteric or bioisosteric replacement [21,22,23,24,25].

It has been claimed that the substitution of the tetrazole ring in candesartan by the structurally identical 5-oxo-1,2,4-oxadiazole ring on azilsartan has led to unique pharmacological properties [26]. 

In an efficient and green synthetic route to losartan, Shangxia et al. [27] in the last step converted the nitrile derivative to tetrazole ring. Their synthetic methodology is devoid of hazardous chemicals like hydrogen chloride and ammonia, making the process convenient and eco-friendly (Scheme 1, [27]).

In a US patent, Reddy A.V. provided the same conversion using a similar reaction (Scheme 2, [28]). The three-step synthetic process uses commercially available 4′-(bromomethyl)-2-cyanobiphenyl (**1**) as a starting material. The main drawback of this synthesis is the use of excessive hazardous organotin reagent in the last step.

### 2.2. Synthetic Routes for Valsartan

Valsartan was approved from the FDA and released to the US market at 1996 under the name of Diovan. Similar synthetic reagents are described by Ghosh et al. who synthesized valsartan that also bears a tetrazole ring (Scheme 3, [29]). In his work, author described three previous synthetic pathways A–C, which all have the same drawback; the use of expensive boronic acid substrates and palladium catalyst in the cross-coupling step. In a refined approach, inexpensive and commercially available o-anisic acid **10** has been used as the starting material.

Zhang et al. (Scheme 4, [30]), Harel et al. (Scheme 5, [31]) and Kankan et al. (Scheme 6, [32]) used nitrile derivatives for cyclization in valsartan. 

Zhang emphasized in the ortho-metalation of *p*-bromotoluene **4** that produces a boronic acid intermediate **10** which was subjected to palladium-catalyzed Suzuki coupling. Implementation of valsartan synthesis using Suzuki reaction is the main advantage of this methodology. The saponification of the methyl ester **11**, was realized in a convenient and economical manner and it is more suitable for industrial production. Overall, the synthesis is characterized by minimum use of expensive and hazardous metals and increased efficiency (Scheme 4).
molecules-26-02927-sch004_Scheme 4Scheme 4Synthesis of valsartan through nitrile derivative 1 [30].
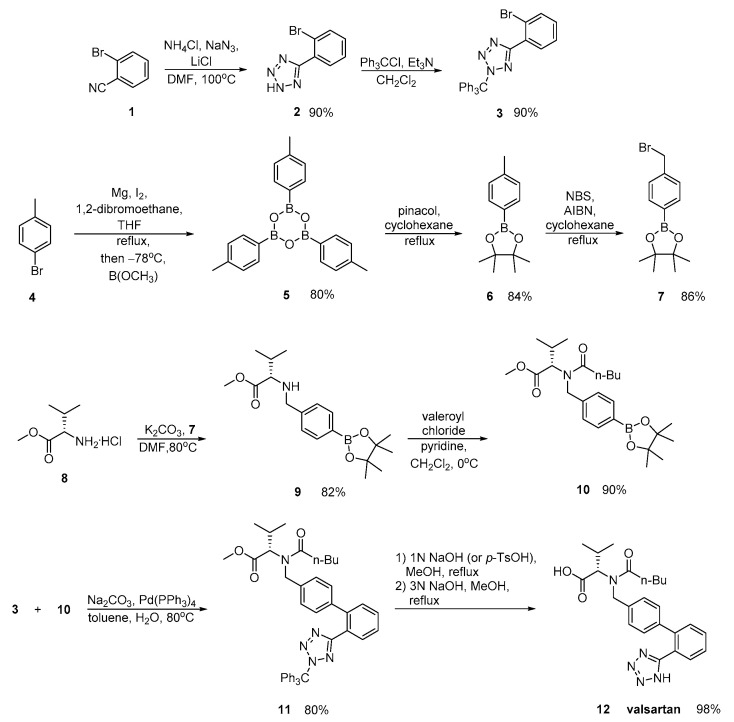


In a patent by Harel and Rukhman, a synthetic process for valsartan is incorporated which is substantially free of its isoleucine analogue. Impurities is a major issue in drug synthesis; therefore, their elimination is the focus of this synthetic methodology (Scheme 5).

**Scheme 5 molecules-26-02927-sch005:**
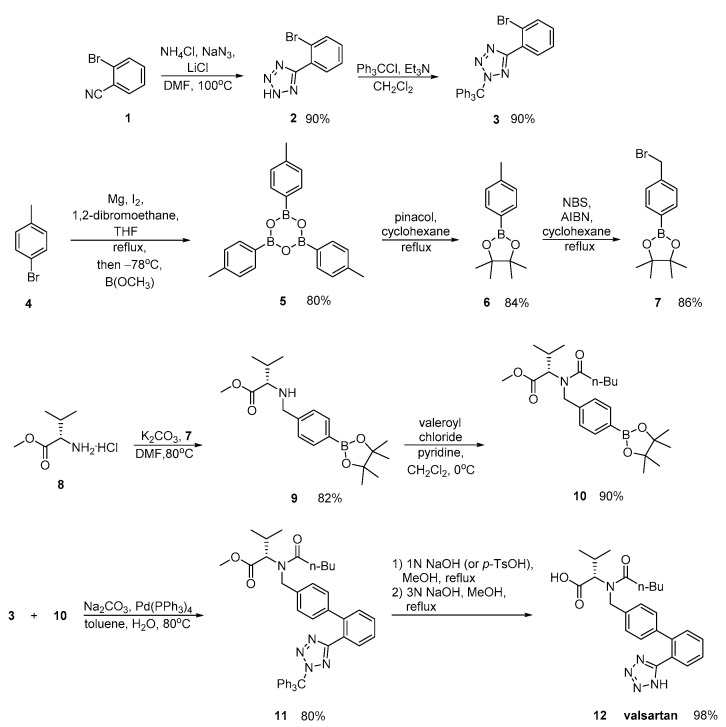
Synthesis of valsartan through nitrile derivative 7 [31].

In a later patent by Kankan et al., an alternative synthesis was described (Scheme 6). The main disadvantages of this synthesis are: the use of toxic tributyl azide, the formation of explosive hydrogen azide, the oily nature of the intermediates which require repeated step of crystallization making the process laborious and time consuming for industrial scale.

**Scheme 6 molecules-26-02927-sch006:**
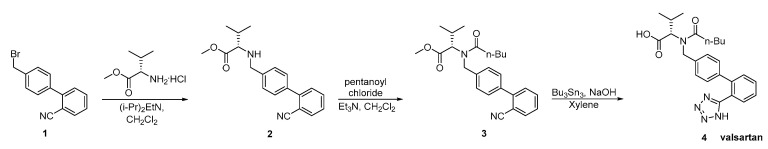
Synthesis of valsartan through nitrile derivative 3 [32].

### 2.3. Synthetic Routes of Irbesartan

Irbesartan was approved from the FDA and released to the US market at 1996 under the name of Avapro. A similar strategy was used for the synthesis of irbesartan (Scheme 7 and Scheme 8, [33,34,35]). In particular, irbesartan was synthesized through dehydration and tetrazole formation in one step from a substituted cyclopentane derivative **6** with tributyltin chloride and sodium azide. The purity of the compound was 99.85% as confirmed by HPLC analysis. Minimization of the impurities was the mail focus of this synthetic methodology (Scheme 7, [33,34]).

In an elegant approach, irbesartan was synthesized through a new catalytic system based on palladium-amido-*N*-heterocyclic carbenes for Suzuki–Miyaura coupling reactions of heteroaryl bromides. In this synthetic method, the biaryl structure was formed using a very low catalytic load making the procedure industrial friendly. Thus, the use of the industry-advantageous Suzuki reaction is the hallmark of this synthetic methodology (Scheme 8, [35]).

**Scheme 8 molecules-26-02927-sch008:**
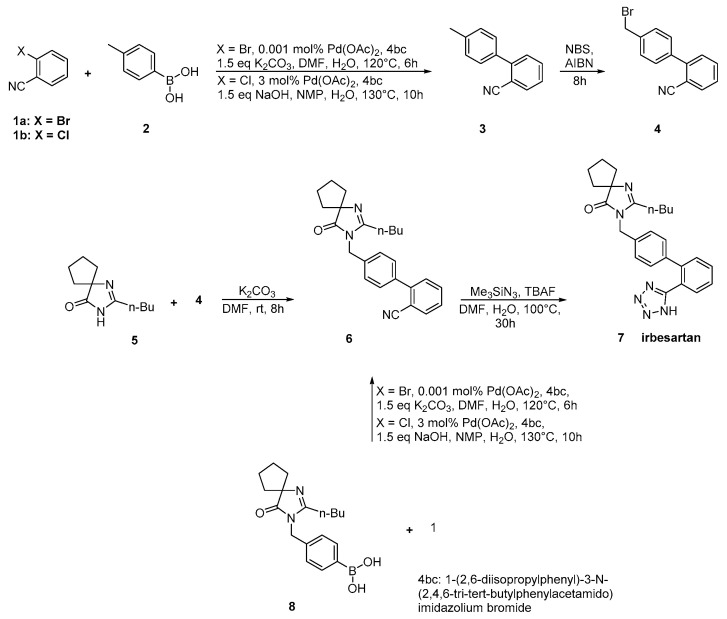
Synthesis of irbesartan through nitrile derivative **6** by Kumar, A.S et al. [35].

### 2.4. Synthetic Routes of Azilsartan

Azilsartan was approved from the FDA and released to the US market at 2011 under the name of Edarbi. Nitriles are also useful for the synthesis of 1,2,4-oxadiazol ring of azilsartan (Scheme 9 and Scheme 10, [36,37]). The method described in Scheme 10, [36] is superior to previous published (Scheme 9, [37]) because it is very efficient (90% yield) and does control the impurities (99.93% HPLC purity). 

It must be pointed out that impurities in the synthesis of AT1R antagonists is an important issue. It is suffice to say that recently Princeton Pharmaceuticals recalled numerous irbesartan tablets and seven lots of irbesartan hydrochlorothiazide (HCTZ) because after testing revealed the drugs contained trace amounts of a carcinogen [38,39].

**Scheme 9 molecules-26-02927-sch009:**
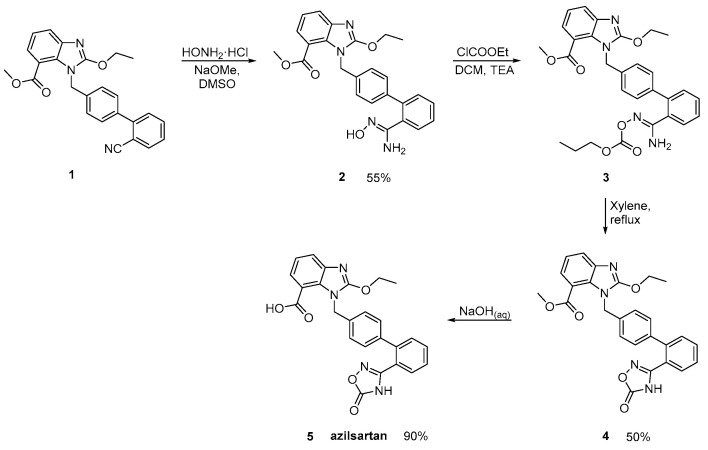
Synthesis of azilsartan. The nitrile derivative **1** is cyclized to final product **5** (azilsartan) [37].

**Scheme 10 molecules-26-02927-sch010:**
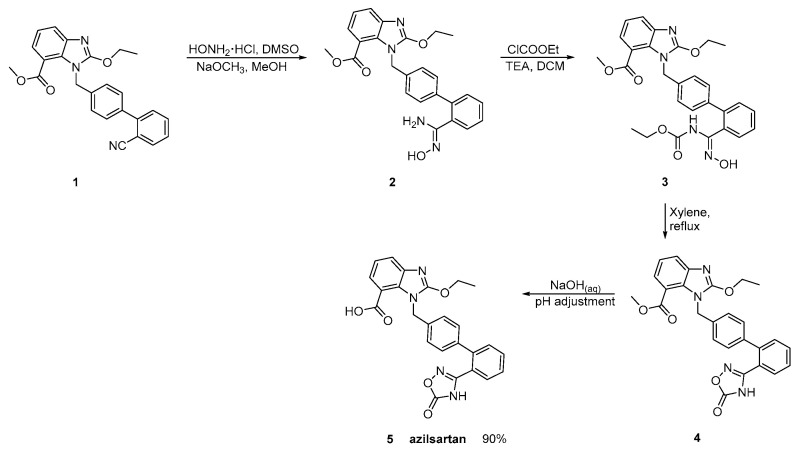
Synthesis of azilsartan. Improved synthesis of the nitrile derivative **1** is cyclized to final product **5** (azilsartan) [36].

### 2.5. Synthetic Routes of Telmisartan

Telmisartan was approved from the FDA at 1998 and released to the US market at 2000 under the name of Micardis. Telmisartan is assembled by two benzimidazole subunits and a biphenyl-2-carboxylic acid segment. A nitrile group can also be used to produce the carboxylate group of telmisartan through hydrolysis. The benzimidazole moiety can be created through an improved, copper-catalyzed, cyclisation of *o*-haloarylamidines obtained from *o*-haloarylamines. The total synthesis started from the commercially available 4-Nitro-*m*-toluic acid and it was achieved in seven steps with an overall yield of 54%. The advantage of this synthesis, relatively to the previous reported, is the absence of the use of possibly harmful HNO_3_/H_2_SO_4_ for nitration and polyphosphoric acid (PPA) for cyclisation (Scheme 11, [40]). 

Telmisartan has also been prepared through different methodologies where the dimer benzimidazole rings are incorporated (Scheme 12, Scheme 13, Scheme 14, Scheme 15 and Scheme 16, [14,41,42,43,44]).

The original synthesis of telmisartan was developed by Ries et al. in 1993 (Scheme 12, [14]), beginning with the stepwise construction of the central benzimidazole ring from 4-amino-*m*-toluic acid methyl ester **1**. Saponification of the resulting substituted benzimidazole **2** was followed by condensation with *N*-methyl-1,2-phenylenediamine using polyphosphoric acid at elevated temperature (150 °C) to afford the functionalized dibenzimidazole **3**. Alkylation of the latter with 4′-(bromomethyl)-2-biphenylcarboxylic acid tert-butyl ester **4** followed by hydrolysis of the resulting ester provided telmisartan in 21% overall yield over eight linear steps. Albeit the relatively low overall yield, the synthetic methodology is attractive because of the reactions’ simplicity and the availability of the cheap reagents.

**Scheme 12 molecules-26-02927-sch012:**
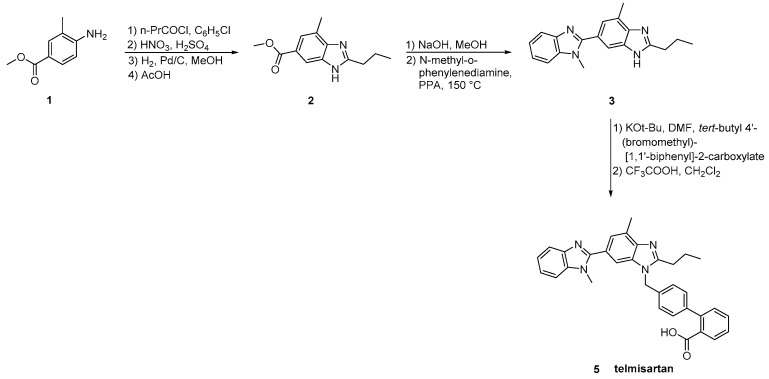
First reported synthesis of telmisartan by Uwe J. Ries et al. [14].

Goossen et al. (Scheme 13, [41]) introduced the biaryl moiety of telmisartan using a Pd/Cu decarboxylative catalyst system. Construction of the biaryl moiety with the aid of a powerful tool like decarboxylative coupling instead of using appropriate reagents containing it is the characteristic of this synthetic pathway.

**Scheme 13 molecules-26-02927-sch013:**
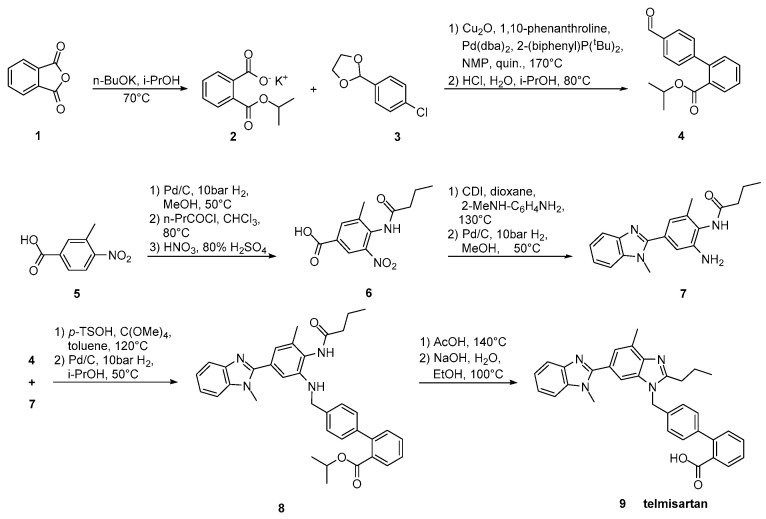
Synthesis of telmisartan by Lukas J. Goossen et al. via decarboxylative cross-coupling [41].

In the method reported by Kumar et al. (Scheme 14, [44]), the biaryl moiety of telmisartan was elaborated by the directed metalation of an appropriate carboxy protecting group followed by transmetalation with zinc chloride and then transition metal catalyzed cross coupling with aryl bromides. Construction of the biaryl moiety is again the hallmark of this synthesis.

**Scheme 14 molecules-26-02927-sch014:**
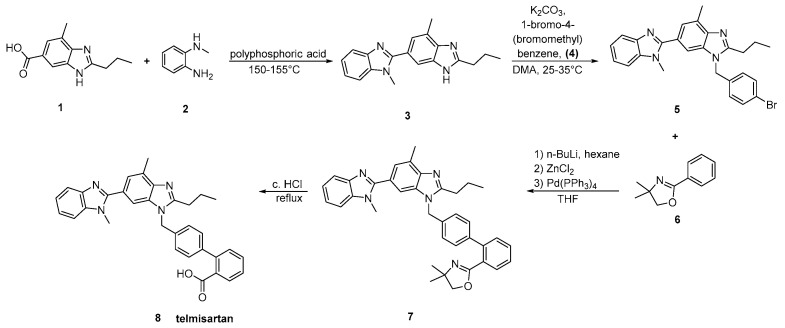
A modified synthesis of telmisartan by A. Sanjeev Kumar et al. [44].

The key strategy in the synthesis reported by Wang et al. (Scheme 15, [42]) was the construction of bis-benzimidazole **7** by reductive cyclization of *o*-nitroaniline **6** with butyl aldehyde. The overall process is operational simple, with low production cost.

**Scheme 15 molecules-26-02927-sch015:**
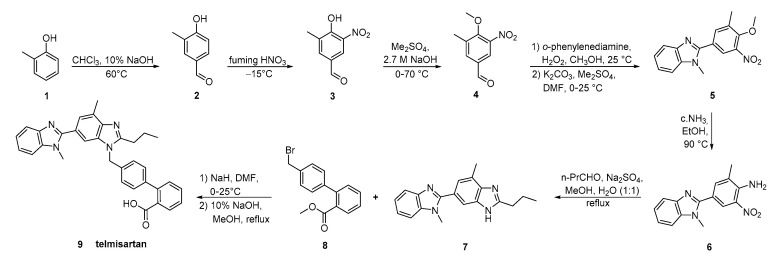
A highly practical and cost-effective synthesis of telmisartan by Ping Wang et al. [42].

In the synthesis reported by Martin et al. (Scheme 16, [43]), two differentially substituted benzimidazole derivatives and a biphenyl-2-carboxylic acid synthon were assembled via a Suzuki cross-coupling reaction, providing telmisartan in 72% overall yield. High overall yielding synthetic methodology makes this procedure attractive.

**Scheme 16 molecules-26-02927-sch016:**
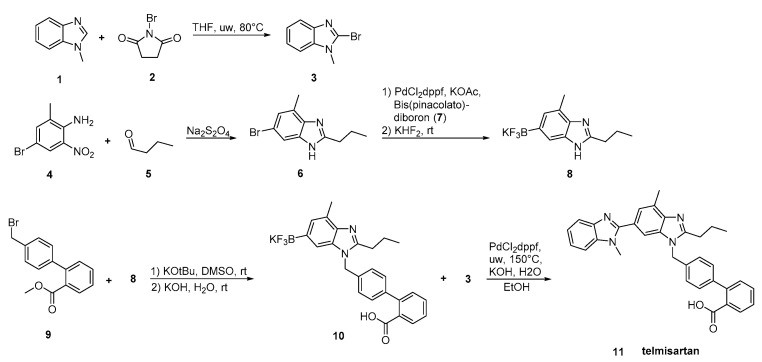
Total synthesis of telmisartan by Alex D. Martin et al. via a Suzuki cross-coupling reaction between two functionalized benzimidazoles [43].

### 2.6. Synthetic Routes of Eprosartan

Eprosartan was approved from the FDA at 2001 and released to the US market at 2003 under the name of Teveten. Eprosartan is one of the AT1R antagonists that does not contain a biphenyl tetrazole ring. It contains two heterocycles, the imidazole and the unique thiophene. Below are reported synthetic routes used to develop it (Scheme 17, Scheme 18, Scheme 19 and Scheme 20, [45,46,47,48,49]).

The process described by Ramakrishnan et al. (Scheme 18, [46]) comprises condensation of valerimidine methyl ester with dihydroxyacetone to give a diacetate which was treated with α-carboxymethylbenzyl alcohol in the presence of triflic acid to give 2-n-butyl-5-acetoxymethyll-(4-carboxyphenyl)-methyl-lH-imidazole. Thus obtained compound on further oxidization with manganese dioxide and thereafter condensation with methyl-3-(3-(2-theinyl)-propionate in the presence of n-butyl lithium at −78 °C gives an ester which is hydrolyzed to give eprosartan. Eprosartan is further converted to its desired salt form.

**Scheme 18 molecules-26-02927-sch018:**
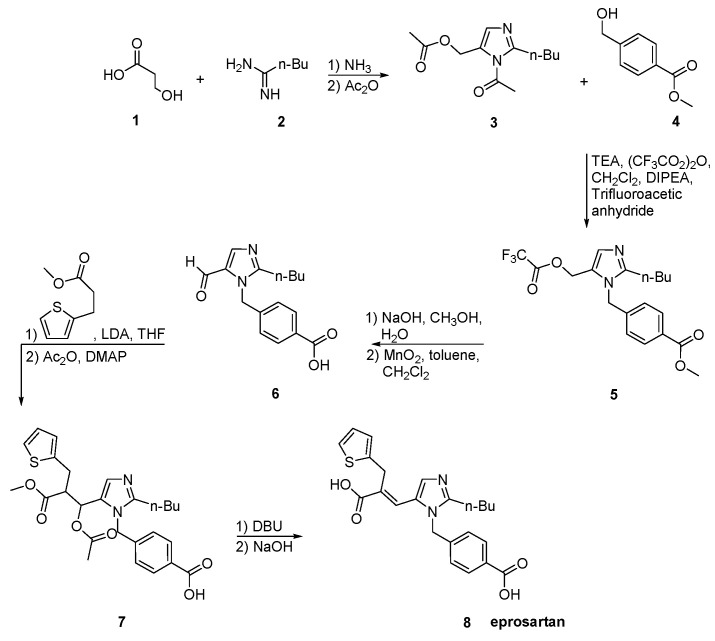
Synthesis of eprosartan by Ramakrishnan et al. [46].

Eprosartan synthesis was reported by Matsuoka et al. to proceed in three stages (Scheme 19, [45]). These stages are: the regioselective protection of 2-n-butyl-4-formylimidazole **1**, followed by reaction with (2-thienyimethyl)-propanedioic acid, mono-alkyl ester and 4-(bromomethyl)benzoate. The efficiency of this synthetic sequence and the quality and yield of eprosartan are particularly important.

**Scheme 19 molecules-26-02927-sch019:**
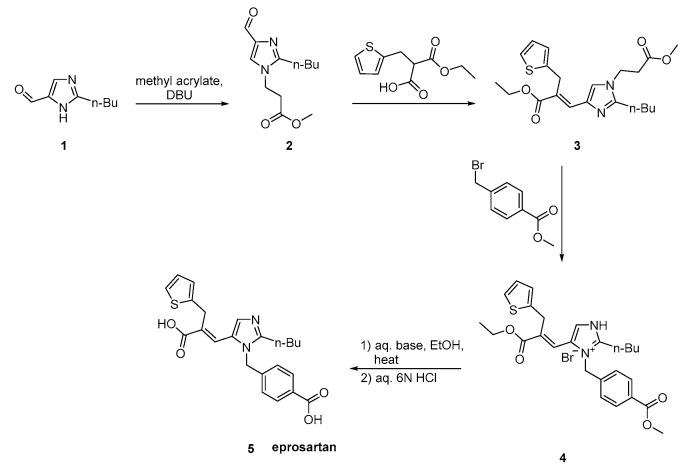
Synthesis of eprosartan by Matsuoka et al. [45].

The synthesis reported by Weinstock et al. starts with alkylation of 4-chloro-5-formyl-2-butylimidazole **1** with *p*-carbomethoxybenzylbromide **2** to give eprosartan in three steps (Scheme 20, [47,48]).

**Scheme 20 molecules-26-02927-sch020:**
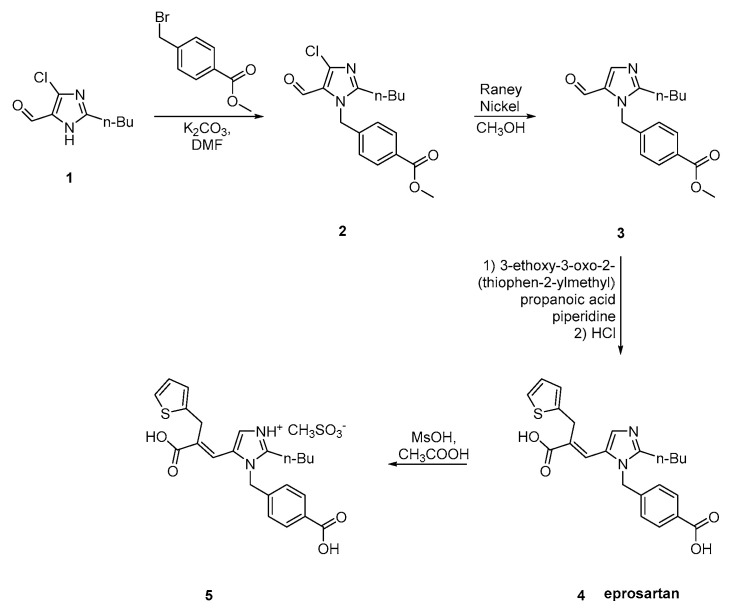
Synthesis of eprosartan by Weinstock et al. [47,48].

### 2.7. Synthetic Routes of Olmesartan Medoxomil & Candesartan Cilexetil

Olmesartan and candesartan are given as pro-drugs olmesartan medoxomil and candesartan cilexetil. Olmesartan was approved from the FDA and released to the US market at 2002 under the name of Benicar while candesartan was approved at 1998 and released to the US market at 1999 under the name of Atacand. Synthetic routes are provided in the following schemes (Scheme 21, Scheme 22, Scheme 23 and Scheme 24, [50,51,52,53]).

Candesartan was synthesized using an efficient protocol for C–H arylation by a catalytic system involving [RuCl_2_(p-cymene)]_2_ and triphenylphosphine (Scheme 21, [50]).

**Scheme 21 molecules-26-02927-sch021:**
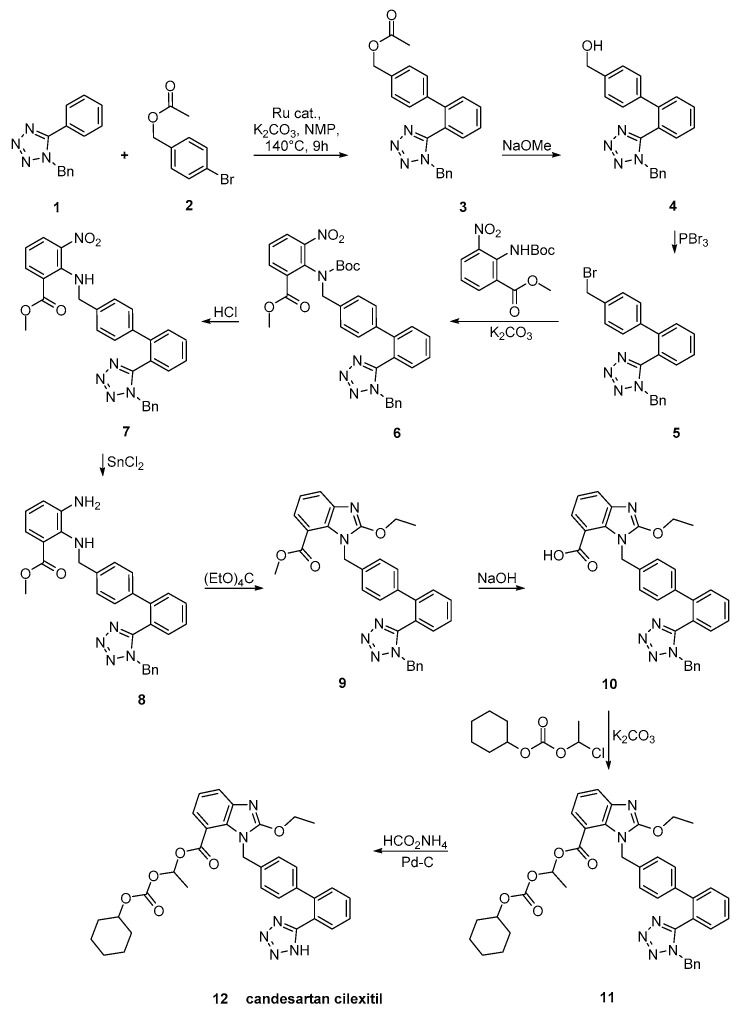
A practical synthesis of prodrug candesartan cilexitil by Masahiko Seki et al. [50].

By careful monitoring of the reaction conditions, Babu et al., described an olmesartan medoxomil synthesis, where impurities were minimized (Scheme 22, [51]).

**Scheme 22 molecules-26-02927-sch022:**
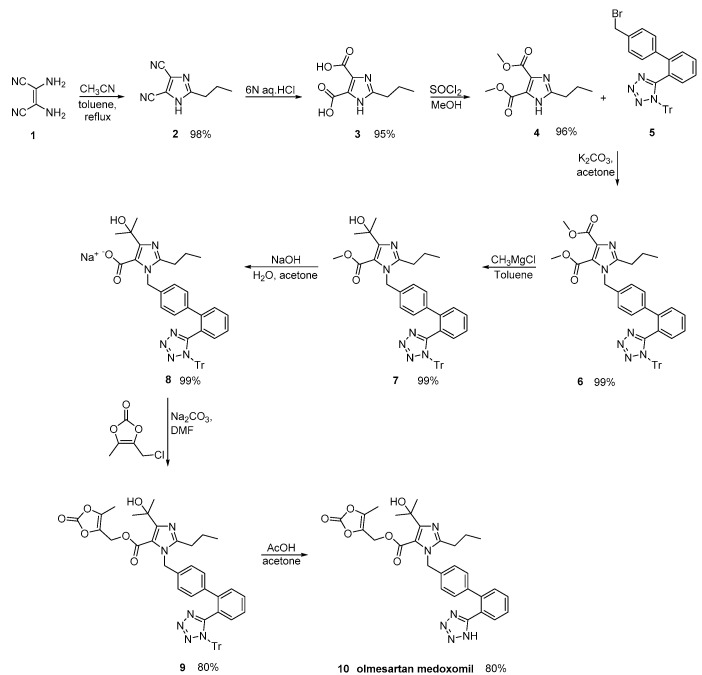
Synthesis of the prodrug olmesartan medoxomil by Babu et al. [51].

Venkanna et al. identified the impurities resulting from the olmesartan medoxomil synthetic process (Scheme 23, [52]).

**Scheme 23 molecules-26-02927-sch023:**
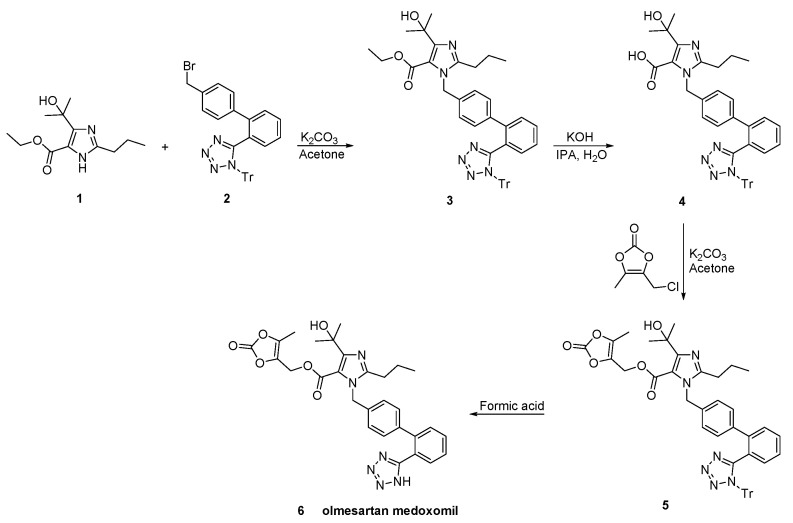
Synthesis of the prodrug olmesartan medoxomil by Venkanna et al. [52].

In a patent filled by Toplak Casar [53], two variations of an one-pot trityl olmesartan medoxomil synthesis have been described (Scheme 24 and Scheme 25). In this methodology, the intermediates are not isolated in each step and by using the same solvent and the same base, no material is removed or exchanged. This feature renders this methodology particularly attractive.

**Scheme 24 molecules-26-02927-sch024:**
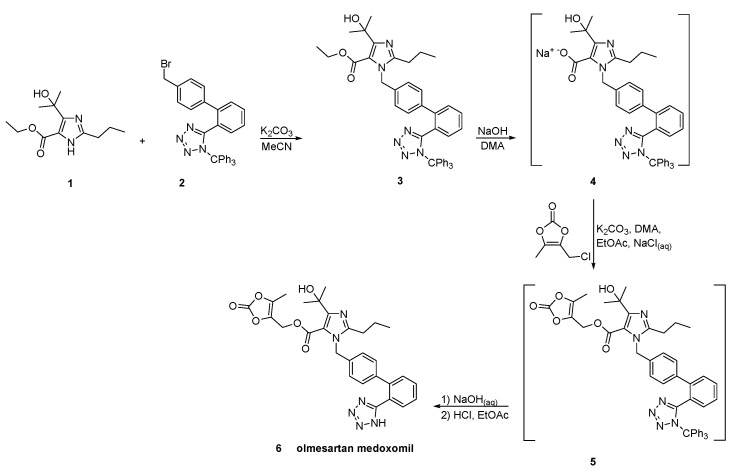
First variation of one-pot synthesis of the prodrug olmesartan medoxomil [53].

**Scheme 25 molecules-26-02927-sch025:**
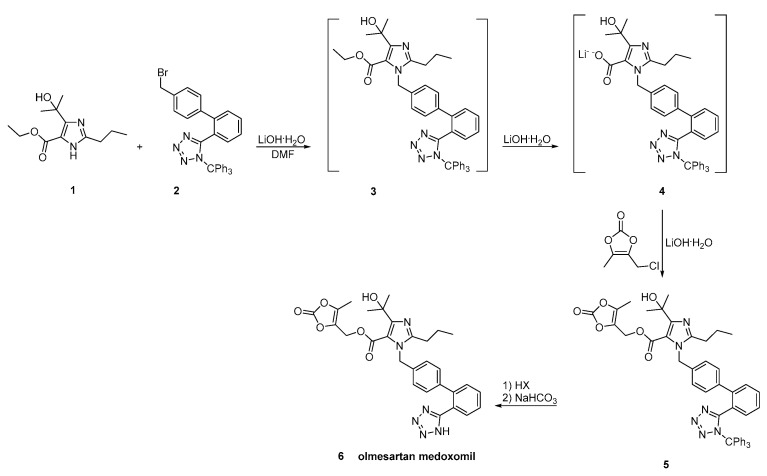
Second variation of one-pot synthesis of the prodrug olmesartan medoxomil [53].

## 3. Conclusions

The development of sartan AT1R antagonists have shaped the basis to compact hypertension. Intriguingly, their rational design has been based on pharmacophore mimicry to the C-terminus of a low energy conformer of the natural hormone AII. Structurally these molecules contain a heterocycle, an alkyl chain, and the majority of them constitute a biphenyl tetrazole segment or an acidic group. Heterocycles such as tetrazole and 1,2,4-oxadiazol ring are derived from nitrile derivatives. The reagents for their preparations are described in the total synthesis of sartans containing these rings. The carboxylate group of telmisartan is also derived from a nitrile derivative. Moreover, candesartan and olmesartan are provided as prodrugs, thus in the form of candesartan cilexitil and olmesartan medoxomil and their synthetic routes are provided.

Another issue described in the total synthesis of sartans is the elimination of impurities. Novel strategies have been invented to eliminate the impurity content that can be detrimental to the human health. Purer, safer and efficient synthesis with lesser steps will be continuing to be a constant challenge for synthetic chemists. This will suppress the recall of danger lots by the pharmaceutical companies to protect human health. 

The synthetic strategy of sartans basically involves two critical synthetic steps: (a) coupling of aromatic derivatives with heterocyclic segments and (b) conversion of nitrile group to an acidic bioisosteric one. The synthetic effort as it is already pointed out is targeting towards the reduction of the original synthetic steps to restrain impurities and enhance reaction yields. There is no doubt that synthetic strategies described herein and involved in the synthesis of sartans could be adapted and re-purposed for other drug molecules possessing common heterocycles. 

## Data Availability

All data, belongs to this work, is given and presented herein.

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
