# Peer review of "Rational Design and Synthesis of AT1R Antagonists"

_molecules, 2021, doi:10.3390/molecules26102927_

Round 1

Reviewer 1 Report

The work of reviewing the summaries and possible alternatives is extremely interesting.

The summaries must be better described in the text in their strength and advantage over the alternatives proposed, otherwise the simply summary list make it ineffective.

The work can be accepted after careful revision of the stylistic and typing errors listed below, and especially after careful improvement of the description.

Conclusions need to be broadened and more attractive considerations need to be made

  • Pay close attention to the use of spaces after punctuation especially in the reaction conditions written on the arrows in the reactions of the schemes and between text and reference.
  • use the words o-, m-, p- to indicate the positions of substituents on the aromatics always in italics spaced by a hyphen and uniform in the patterns and text. Pay attention to this position in the Scheme 13
  • pay attention to the stoichiometric coefficients in the formulas, always write them in subscript (scheme 15...)
  • pay attention to the numbers of the compounds, always in bold also in the captions.
  • Write always: (Scheme xx, [nr]). See line 235...
  • Use only brackets and non-round brackets for bibliographical reference numbers (line 166, 248,252)
  • Put the yield always to the right of the number of the referring compound (Scheme 9).
  • Use comma to separate reagents on arrows in synthesis schemes (Scheme 11)
  • Rewrite some sentences (line 184)
  • Do not use the bold for caption (Scheme 10)
  • Put the name of compound and its number close and aligned (see in Scheme 11, 12…)
  • Verify the structures of compounds 4 and 7 in Scheme 13.
  • Briefly describe the synthesis 12-16, 17-20. 21-23, 24-25.
  • Use et al. always (Scheme 17, 18, 20…; lines 241)
  • Use capital letter or not for common names (line 273)
  • Line 127: is a separate scheme?
  • Insert the names of authors or the corresponding references in the captions to differentiate the routes.

Reviewer 2 Report

The Renin Angiotensin System (RAS) discovery unveiled a path to develop efficient drugs to combat hypertension efficiently. Several compounds that prevent the Angiotensin II hormone from binding and activating the AT1R, named sartans, have been developed. In this paper, the authors reported a review of the synthetic paths followed for developing different sartans since the discovery of losartan.

Although I recommend the publication of this paper after minor revisions, some comments and suggestions could be taken into account by the authors.

  • The biggest drawback of the article is the schemes. I consider that they should redo them in such a way that they help to a better understanding, but also without blank spaces that do not contribute.
  • On the other hand, the authors described the synthetic paths followed for developing different sartans. However, no discussion is evident or at least some type of comparison between the different sartans. In my opinion, some discussion or at least some type of comparison between the different sartans synthetic paths would be advisable.
